# CCL21 Induces Plasmacytoid Dendritic Cell Migration and Activation in a Mouse Model of Glioblastoma

**DOI:** 10.3390/cancers16203459

**Published:** 2024-10-12

**Authors:** Lei Zhao, Jack Shireman, Samantha Probelsky, Bailey Rigg, Xiaohu Wang, Wei X. Huff, Jae H. Kwon, Mahua Dey

**Affiliations:** 1Department of Neurosurgery, University of Wisconsin School of Medicine & Public Health, UW Carbone Cancer Center, Madison, WI 53706, USA; lzhao73@wisc.edu (L.Z.); jshireman@wisc.edu (J.S.); probelsky@wisc.edu (S.P.); brigg@wisc.edu (B.R.); wang@neurosurgery.wisc.edu (X.W.); 2Department of Neurological Surgery, Indiana University School of Medicine, Indianapolis, IN 46202, USA; wxia@iupui.edu (W.X.H.); kwonjaeh@iu.edu (J.H.K.)

**Keywords:** glioblastoma, CCL21, plasmacytoid dendritic cells, CCR7, immunosuppression

## Abstract

**Simple Summary:**

Glioblastoma is known to be highly immunosuppressive in nature. Plasmacytoid dendritic cell infiltration is known to be associated with glioblastoma progression and promotes tumor immunosuppression. We provide important details to indicate that glioblastoma cell-secreted CCL21 plays a dual role both in recruiting plasmacytoid dendritic cells via binding to CCR7 and activating plasmacytoid dendritic cells through the CCR7/ACKR4—β-arrestin/CIITA pathway. Our result also provides a rationale to therapeutically target CCL21 as a potential novel treatment for glioblastoma.

**Abstract:**

Dendritic cells (DCs) are professional antigen-presenting cells that are traditionally divided into two distinct subsets: myeloid DCs (mDCs) and plasmacytoid DCs (pDCs). pDCs are known for their ability to secrete large amounts of cytokine type I interferons (IFN- α). In our previous work, we have demonstrated that pDC infiltration promotes glioblastoma (GBM) tumor immunosuppression through decreased IFN-α secretion via TLR-9 signaling and increased suppressive function of regulatory T cells (Tregs) via increased IL-10 secretion, resulting in poor overall outcomes in mouse models of GBM. Further dissecting the overall mechanism of pDC-mediated GBM immunosuppression, in this study, we identified CCL21 as highly upregulated by multiple GBM cell lines, which recruit pDCs to tumor sites via CCL21-CCR7 signaling. Furthermore, pDCs are activated by CCL21 in the GBM microenvironment through intracellular signaling of β-arrestin and CIITA. Finally, we found that CCL21-treated pDCs directly suppress CD8+ T cell proliferation without affecting regulatory T cells (Tregs) differentiation, which is considered the canonical pathway of immunotolerant regulation. Taken together, our results show that pDCs play a multifaced role in GBM immunosuppression, and CCL21 could be a novel therapeutic target in GBM to overcome pDC-mediated immunosuppression.

## 1. Introduction

Dendritic cells (DCs) are a diverse group of specialized antigen-presenting cells with key roles in the initiation and regulation of innate and adaptive immune responses. Generally, DCs can be classified into two subsets: myeloid DCs (mDCs) or classical DCs (cDCs) and plasmacytoid DCs (pDCs). While mDCs mainly process and present antigens to T cells, pDCs are important mediators of an immune response because of their ability to produce a large amount of the proinflammatory cytokine IFN-α. The function of pDCs is largely dependent on the surrounding environment and the type of stimulation, with both immunogenic and tolerogenic phenotypes possible [1,2,3]. In the context of tumor biology, this functional variability can lead to either pro- or anti-tumor progression.

Glioblastoma (GBM) is the most common and aggressive primary brain tumor in adults, with a median overall survival of around 21 months with the most aggressive standard of care regimen [4], and a 100% recurrence rate. It has been demonstrated that the GBM tumor microenvironment (TME) is extremely immunosuppressive, which largely limits the efficacy of immunotherapy. Previous work from our lab has shown pDCs, which infiltrate GBM, to be associated with tumor progression and poor prognosis. GBM infiltrating pDCs display downregulation of TLR-9 and lose the ability to produce IFN-α. Furthermore, they drive the recruitment of IL10+ regulatory T cells (Tregs), promoting a highly suppressive tumor microenvironment [5]. However, two questions remain. First, how GBM cells recruit pDCs to tumor sites, as it is found that pDCs are significantly increased under pathological conditions [6,7]. Second, how GBM cells alter pDC function that eventually contributes to the overall immunosuppressive TME.

Chemokines are an important bridge between tumor cells and immune cells. On the one hand, chemokines and chemokine receptors regulate immune cell chemotaxis, homeostasis, and tolerance mechanisms [8,9]. On the other hand, a large diversity of chemokine signaling can increase the migration and invasion abilities of tumor cells and mediate tumor cell escape from immune surveillance [10,11,12,13]. CCL21-CCR7 signaling is found to be upregulated in many different types of cancer, including GBM [14,15,16,17]. Several studies have reported that CCL21-CCR7 signaling increases glioma cell invasion and EMT [18], recruits microglia and macrophages [19], and promotes tumor progression [20]. As a key regulator of CCR7, the atypical chemokine receptor 4 (ACKR4) binds to the CCR7 ligands CCL19 and CCL21, as well as the CCR9 ligand CCL25. ACKR4 has also been shown to regulate the migration of DCs into lymphatics as well as T cell areas of LNs [21,22,23,24]. Despite these advances, the role of CCR7/ACKR4 signaling is largely cancer-type dependent, showing either an unfavorable or beneficial role [21,25]. Furthermore, the mechanism of CCR7/ACKR4 signaling in the GBM immunosuppressive microenvironment is still largely unknown.

In this study, using a mouse model of GBM, we identified CCL21 as highly upregulated by GBM cells. GBM recruits pDCs to tumor sites via CCL21-CCR7 signaling. Furthermore, pDCs are activated by CCL21 in the GBM TME through intracellular signaling of β-arrestin and CIITA. Finally, we explored the regulatory function of pDCs induced by CCL21, which might be a potential therapeutic target for GBM.

## 2. Materials and Methods

### 2.1. Cell Culture and DC Isolation

All cells utilized for experimentation were maintained in a sterile culture within a cell incubator (Thermo Fischer, Waltham, MA, USA) at 37 degrees with 5% CO_2_ supplementation. Mouse tumor lines GL261 and CT2A were maintained with DMEM supplemented with 10% fetal bovine serum and 5% penicillin/streptomycin antibiotic. DCs were isolated from 6- to 8-week-old wild-type mouse bone marrow progenitor cells as described previously [26,27]. mDCs were differentiated in RPMI supplemented with IL-4 (10 ng/mL) and GM-CSF (10 ng/mL). pDCs were differentiated in RPMI supplemented with Flt3-L (20 ng/mL). When indicated, cells were treated with CCL21 protein at 100 or 300 ng/mL. Neutralized antibodies anti-CCL21 and anti-CCR7 were added at 50 μg/mL. The reagent list is included in the Appendix A.

Cell lines used in this study were generously provided by collaborators and have been extensively published by our lab and many standard labs in the field [5,27,28,29].

### 2.2. Mice

All animals used in experiments were maintained in accordance with the IRB and IACUC committee rules established at the University of Wisconsin. C57/Bl6 mice used for research were obtained from Charles River laboratories. The ages of the mice used in the experiments are noted specifically within the results section and ranged from 6 to 8 weeks. All mice were contained in a temperature-controlled animal room, maintaining a 12 h light and dark cycle. For tumor implantation, a total of 4 × 10^5^ GL261 cells suspended in 5 μL PBS were intracranially (i.c.) injected into the brains of the mice using a stereotactic frame with coordinates 2 mm behind the bregma suture and 1.5 mm posterior to the midline suture. The skin incision was closed using sterile vicryl sutures, and mice were given post-procedure analgesics (buprenorphine) to ease discomfort, as described previously [5,27]. In all animal experiments, 5–6 mice were randomly grouped, and a straight 50/50 breakdown of males and females was maintained. Neutralized anti-CCL21 antibodies were administrated at 0.5 mg/kg twice a week by i.p. injection [30].

### 2.3. Cytokine/Chemokine Array

Brain tumor tissues were dissociated from three GL261 tumor-bearing mice (3 wk post-intracranial injection) or normal mice. Tissue lysates were pooled together and incubated with the Mouse Cytokine/chemokine Array C6 (AAM-CYT-6, RayBiotech, Peachtree Corners, GA, USA) and developed according to the manufacturer’s protocol. The film was scanned and analyzed for densitometry using Fiji ImageJ 2.15.1.

### 2.4. Immunohistochemistry and Immunofluorescence

Immunohistochemistry was carried out on 8-micron sections of frozen brains after an overnight soak in a sucrose solution. Sections were mounted to cover glass, stained with hematoxylin and eosin, and then sealed for brightfield imaging using a Keyence microscope. For immunofluorescence on tumor cells and DCs, cells were seeded on coverslips and fixed in 4% PFA. The coverslips were blocked using goat serum and then stained with antibodies (antibodies list included in the Appendix A) overnight. After overnight staining, secondary antibodies were stained for 1 h at room temperature. The coverslips were mounted for visualization on a Keyence fluorescence microscope.

For CCL21 immunofluorescent staining, although CCL21 as a chemokine could be secreted extracellularly, the cells were washed intensely during staining to wash out extracellular CCL21 immunofluorescence to capture true intracellular staining.

### 2.5. Migration Assay

For Transwell migration, 105 cells in 100 μL RPMI medium were seeded in the upper chambers of 24-well tissue culture plates with 5 mm pore-size inserts (Corning, NY, USA). The lower chambers were loaded with 600 μL medium with attractant and/or neutralized antibodies. GBM cell lines (GL261 and CT2A) were cultured in the lower chambers of 24-well tissue culture plates. Migrating cells were collected from the lower chambers and the bottom of cell inserts after overnight incubation at 37 degrees and quantified by flow cytometry.

For the Incucyte migration assay, 5 × 10^5^ cells were seeded in a 6-well plate in 2 mL RPMI medium with CCL21 protein and/or neutralized antibodies. Cells were stained for BST2 and CD11c. Cell movement was serially photographed every 2 h for 24 h using an Incucyte (Sartorius, Göttingen, Germany) incubator. The resulting image files were analyzed by Fiji ImageJ 2.15.1. Cell movement was analyzed and quantified from the resulting image files by TrackMate (7.14.0)/ Fiji ImageJ 2.15.1. N ≥ 60 cells were analyzed in each condition and plotted in Graphpad Prism 7.0 or Microsoft Excel.

### 2.6. Flow Cytometry

Single-cell suspensions were made from the brain, blood, lymph node, or bone marrow, as described in our previous publication [5,27]. For cellular staining, cells were incubated with related antibodies (the antibody list is included in the Appendix A). For cytokine assay, DCs were stimulated with type A CpG with protein transport inhibitors for 6 h at 37 °C in a humidified atmosphere of 95% air/5% CO_2_. Cells were then fixed and permeabilized overnight at 4 °C using Fix/Perm buffer (eBioscience, San Diego, CA, USA) according to the manufacturer’s instructions and stained with intracellular antibodies. All flow cytometric analyses were performed using an Attune flow cytometer (Thermo Fischer, Waltham, MA, USA). The gating strategy is provided in Appendix A. Flow analysis was conducted in Flowjo 10.10, and results were plotted using Graphpad Prism 7.0. To verify the specificity of the ACKR4 primary antibody, autofluorescence subtraction was performed post-acquisition in Flowjo 10.10 [31]. Briefly, the secondary antibody-only staining was treated as an additional parameter to estimate during compensation. The unstained served as the compensation control.

### 2.7. Real-Time Quantitative PCR Assay

RNA was extracted from cells using RNeasy Kits (Qiagen, Germantown, MD, USA) following the manufacturer’s instructions. Purified mRNA (2 mg) was used for the first-strand cDNA synthesis using an iScript cDNA synthesis kit (Bio-Rad, Hercules, CA, USA), and quantitative RT-PCR was performed using the ABI QuantStudio 5 real-time PCR system (Thermo Fischer, Waltham, MA, USA). Primers are detailed in Appendix A. Each cDNA template was amplified in triplicate using SYBR Green PCR Master Mix (Bio-Rad, Hercules, CA, USA). Primer sequences and cycle numbers are included in the Appendix A.

### 2.8. Western Blot

Nuclear and cytoplasm proteins were extracted using a nuclear extraction kit according to the manufacturer’s instructions (Abcam, Cambridge, UK). Protein concentrations were determined using a Bio-Rad DC Protein Assay kit. Approximately 3–10 μg of proteins from each sample were separated on 4–20% Mini-PROTEAN TGX precast gels (Bio-Rad, Hercules, CA, USA) and transferred to PVDF membranes. Proteins of interest were detected by immunoblotting using the primary and secondary antibodies detailed in the Appendix A. Specific protein bands on the blots were visualized by applying enhanced chemiluminescence reagents according to the manufacturer’s instructions (Pierce, Rockford, IL, USA) and then recorded with a LAS-4000 Mini imager (GE, Piscataway, NJ, USA). The densities of the protein bands were analyzed and quantified by Fiji ImageJ 2.15.1.

### 2.9. In Vitro Proliferation Assay

Naive CD4+ T cells were isolated from the spleen of OT-II mice using a mouse Naive CD4+ T Cell Isolation Kit (MiltenyiBiotec, Bergisch Gladbach, Germany). CD8+ T cells were isolated from the spleen of WT mice using a mouse CD8+ T Cell Isolation Kit (MiltenyiBiotec, Bergisch Gladbach, Germany). Cells were labeled with CellTrace Violet according to the manufacturer’s instructions (CellTrace Violet proliferation kit, Thermo Fischer, Waltham, MA, USA) and used as responder cells. pDCs and mDCs were separated using a mouse Plasmacytoid Dendritic Cell Isolation Kit (MiltenyiBiotec, Bergisch Gladbach, Germany) and flow sorting. For antigen-specific stimulation of CD4+ T cells isolated from OT-II mice, pDCs and mDCs were preloaded with OVA323-339 peptides (30 mg/mL, InvivoGen, San Diego, CA, USA) for 3 h. Non-specific stimulation was carried out with anti-CD3 (145-2C11; 0.5 mg/mL), and after 72 h at 37 °C, proliferation was determined by flow cytometry.

### 2.10. Statistics

Statistically significant differences between treatment groups were determined by one-way ANOVA (Graphpad Prism 7.0, La Jolla, CA, USA) using the Bonferroni multiple comparison post hoc test or a two-tailed t-test for grouped comparison. * *p* < 0.05, ** *p* < 0.01, and *** *p* < 0.001 were used to show statistical significance throughout the article. Error bars in histograms show standard deviations of triplicate measurements.

## 3. Results

### 3.1. CCL21 Is Highly Expressed by Mouse GBM Cells and Associated with Decreased GBM Patient Survival

As chemokines play an essential role in DC function, we hypothesized that GBM cells recruit and affect pDCs through chemokine/cytokine production and secretion. To test this hypothesis, we performed an unbiased mouse chemokine/cytokine array from tumor-bearing brain (GL261) and normal brain tissue. This showed that CCL21 is highly produced and secreted in the tumor brain (mean intensity of 14,183.38 ± 775.56 for the tumor brain versus 7682.11 ± 773.18 for the normal brain, *p* < 0.001, Figure 1A), which is consistent with the immunohistochemistry staining of CCL21 protein expression in GL261-bearing brain and normal brain tissue (Figure 1B). Furthermore, we confirmed the intracellular expression of CCL21 in two different mouse GBM cell lines, GL261 and CT2A, by immunocytochemistry staining (Figure 1C). CCL21 is the functional ligand for C-C chemokine receptor 7 (CCR7), a chemokine receptor that plays a pivotal role in guiding T cell migration in health and disease [11,32]. In addition to T cell migration, the CCL21/CCR7 axis also plays an important role in other immune cell trafficking [16,17,33].

To determine the clinical relevance of CCL21 in the context of GBM, we obtained gene expression data and clinical parameters from GBM patient data using The Cancer Genome Atlas (TCGA). The RNA sequencing database from a GBM patient cohort (156 patients, four non-tumor controls) showed a trend toward upregulated expression of CCL21 (*p* = 0.22, Appendix A). Moreover, comparing GBM patients with CCL21-high expression to CCL21-low expression groups, we found that patients with CCL21-low expression had significantly higher median survival compared to the CCL21-high subset (Log-rank *p* = 0.0242, Wilcoxon *p* = 0.0491, Appendix A). These data indicate that increased CCL21 expression in GBM might be one of the factors contributing to overall poor patient survival.

### 3.2. GBM Mediates Preferentially pDC Migration via CCL21

Under pathological conditions such as a tumor or infection, DCs migrate to the brain and spinal cord through either lymph nodes or the vasculature [6,34]. Our previous data demonstrated that pDCs were highly infiltrative into the GBM TME in patients’ tumors as well as mouse models of GBM [5] and our unbiased chemokine/cytokine screen showed significant upregulation of CCL21 in the TME. Because of this, we hypothesized that CCL21, secreted by the GBM cells, drives pDC recruitment to the tumor. In line with our previous report [5], for this study, BST-2 was used as a marker to differentiate mice pDCs and mDCs. Bone marrow-isolated DCs were split between CD11c+BST-2+ pDCs and CD11c+BST-2- mDCs (Figure 2A and Appendix A). The purity of the pDC population was further verified by flow gating of CD11c+BST-2+Siglec-H+ (Appendix A).

To determine the migration capacity in response to the CCL21 protein, a transwell migration assay was employed (Figure 2A). Preferential DC migration was determined by subjecting pooled bone marrow-derived pDCs and mDCs to migrate through the transwell and the subsequent analysis of the migrated DCs with flow cytometry gating on pDC and mDC populations. The result showed statistically significant dose-dependent migration of pDCs in response to CCL21 protein (Figure 2B). We also checked DC transwell migration in the setting of GBM cells and DC co-culture and found that preferentially, pDCs migrated to GBM cells in culture (GL261 and CT2A). There was no comparable difference in migration efficacy between CCL21 protein and tumor co-culture conditions. Interestingly, this migration is not observed in the mDC population, using either CCL21 protein or GBM cells in co-culture. The main receptor of CCL21 and CCR7 signaling plays an essential role in DC maturation and migration [8]. So, we used CCL21 or CCR7 antibodies to block the migration of pDCs in response to CCL21 protein or tumor co-culture conditions. We found that pDC migration can be blocked by either CCL21 or CCR7 antibody neutralization, which suggests a sufficient and necessary regulation of CCL21-CCR7 signaling. To further evaluate the contribution of ACKR4 to pDC migration, ACKR4+ and ACKR4+/CCR7+ pDCs were sorted from bone marrow-derived pDCs and migrated through the transwell in response to CCL21 protein. We found significantly higher migration in the ACKR4+/CCR7+ subset compared to the ACKR4+ alone subset. The result suggests an important role of CCR7 in pDC migration (Figure 2C). Interestingly, we did not find a big population of CCR7+ pDCs, implying that CCR7 expression is dependent on pDC activation (ACKR4+). This was verified by our in vivo data from mouse brain tumors (Appendix A).

Since the effect shown by the transwell migration assay was due to chemotaxis (directed cell movement in response to a chemokine gradient), we also want to check if chemokinetic (random cell movement) activity was also affected by CCL21 protein [35]. We performed the IncuCyte live cell analysis at 2 h intervals over a 24 h period to visualize and quantify random cell movement (Figure 2D–F and Appendix A). The result revealed a statistically significant increased radial movement of pDCs, but not mDCs, to CCL21 protein treatment. Similarly, this migration can also be blocked by either CCL21 or CCR7 antibody neutralization. Taken together, our results demonstrated that GBM mediates pDC migration through the CCL21-CCR7 signaling axis.

### 3.3. GBM Mediates pDC Activation via CCL21

After establishing that CCL21 directly affects the migration of pDCs, we next sought to understand whether CCL21 changes the phenotype of pDCs. pDCs from tumor-bearing mouse brain, blood, lymph node (LN), and bone marrow (BM) one-week post-intracranial tumor implantation were analyzed for the presence of activating markers (MHC-II and CIITA) and compared between tumor-bearing (red) and normal (black) mice (Figure 3A,B). Consistent with our previous study, significant high infiltration of pDCs was found in the tumor-bearing brain (tumor vs. normal: 35.4% vs. 9.4%, *p* < 0.05), and nearly 100% of them (98.5%) were MHC-II and CIITA double-positive. In the LN, BM, and blood from tumor-bearing mice, fewer pDCs were found (tumor vs. normal: blood 16.2% vs. 7.1%; LN 19.1% vs. 8.8% *p* < 0.05, BM 23.2% vs. 9.0% *p* < 0.05). Moreover, there was no consistently significant difference noted in the expression of the activation markers of pDCs from the LN, BM, and blood.

As a key regulator of CCR7 signaling, the atypical chemokine receptor 4 (ACKR4) binds to the CCR7 ligand CCL21 as well [25]. Although little is known, the main role of ACKR4 is to modulate adaptive immunity by regulating immune cell migration and maturation [25]. We found an increase of ACKR4 in GBM-infiltrated pDCs as well as a significant increase in MHCII/ACKR4 double-positive pDCs in tumor-bearing mice (tumor pDC vs. normal pDC: 79.4% vs. 58.8% *p* < 0.01; tumor pDC vs. tumor mDC: 79.4% vs. 39.3% *p* < 0.01, Figure 3A,C and Appendix A). To further evaluate the contribution of ACKR4 to GBM, pDCs and mDCs were sorted from the spleen, LN, and brain of GL261 tumor-bearing mice. We found significantly increased expression of ACKR4 in GBM-infiltrated pDCs compared to pDCs from spleen/LN and all the mDCs (Figure 3A,D). To test if this observation is specific to GBM or if it has implications in other cancers, we evaluated a publicly available cervical cancer dataset. This result was further confirmed by the single-nucleus RNA sequencing data in cervical cancer [36,37] (Figure 3E). All these data suggest ACKR4 might play a role in the activation of tumor-infiltrating pDCs.

To further confirm the role of CCL21, we utilized an in vitro stimulation assay where both pDCs and mDCs were treated by CCL21 or tumor cell co-culture for 24 h (Figure 3F). Flow cytometric analysis of the cells revealed significantly increased MHC-II and ACKR4 double-positive pDCs compared with mDCs or no treatment control, like our in vivo data. Moreover, both CCL21 and tumor-triggered pDC activation can be blocked by either CCL21 or CCR7 antibody neutralization.

### 3.4. CCL21 Induces pDC Activation through Intracellular Signaling of β-Arrestin and CIITA

Next, we explored the molecular mechanism of CCL21-mediated pDC activation. Our data show, both in vivo and in vitro, that the activation marker, MHC-II expression, was increased in pDCs in the context of GBM. Since MHC-II expression can be regulated at the transcriptional or post-transcriptional level [38], we first checked the mRNA expression of five different mouse MHC-II isoforms. Four out of five isoforms were significantly upregulated in CCL21-treated pDCs compared to the no-treatment control. In mDCs, none of the isoforms were induced by CCL21 (Figure 4A). Moreover, CIITA, the MHC class II master regulator, was also significantly upregulated in CCL21-treated pDCs. As the differential expression of CIITA is driven by various promoters [39], three different pairs of primers [40,41,42] were used to target p1, p3, and p4 promoter-driven isoforms, respectively. Reported as a pDC-specific promoter in a previous study [39], in our study, the p3-driven CIITA isoform was significantly increased in CCL21-induced pDCs.

Previous studies have shown that the recruitment of CIITA to the enhanceosome at the MHC-II promoter is necessary and sufficient to induce MHC-II expression [38]. Upon CCL21 binding with its chemokine receptors, such as CCR7, intracellular GPCR signaling pathways activate, causing β-arrestin to translocate from the cytoplasm to the nucleus and associate with transcription co-factors such as p300 and cAMP-response element-binding protein (CREB) [43,44,45]. To verify this signaling pathway in our model system, we confirmed the nuclear translocation of β-arrestin and CIITA by both immunocytochemistry staining and western blot. We found that in non-treated pDCs and mDCs, β-arrestin and CIITA were distributed in both the nucleus and cytoplasm, while an apparent accumulation of β-arrestin and CIITA in the nucleus was observed after 1 h of CCL21 treatment (Figure 4B,C). Western blot analysis of nuclear and cytoplasm fractions indicated that the concentration of β-arrestin and CIITA in the nucleus was increased by about 50% in pDCs after CCL21 treatment or tumor cells co-culture (Figure 4D,E, Appendix A).

### 3.5. CCL21-Induced pDCs Show a Regulatory Phenotype

To further discriminate the immunogenic or tolerogenic function of CCL21-induced pDCs, we first tested the phenotype of CCL21-induced pDCs by flow cytometry. The activating markers MHC-II and CD80 were increased in CCL21-induced pDCs, indicating an activated phenotype. Interestingly, the inhibitory marker PD-L1 was also increased upon CCL21 treatment (Figure 5A).

In response to CpG stimulation, pDCs are known to produce large amounts of IFN-a via TLR-9 signaling [46,47]. In our study, we found that in CCL21-untreated pDCs, both TLR9 and IFN-a are increased upon CpG stimulation. However, CpG-mediated TLR9 expression and IFN-a production were impaired in CCL21-treated pDCs. Instead, when stimulated with CpG’s, CCL21-treated pDCs produced a significantly higher amount of inhibitory cytokine IL-10 (Figure 5B). As important regulators of TLR-9 signaling, IRF7 and MyD88 play vital roles in IFN-a gene transcription in pDCs [48]. In our study, we found both genes showed increased expression in untreated pDCs upon CpG stimulation; however, consistent with the IFN-a data, they were impaired in the presence of CCL21 treatment (Appendix A).

These results suggest a regulatory phenotype of CCL21-induced pDCs, which may contribute to the broad immunosuppressive TME seen in GBM. To confirm this hypothesis, we tested whether CCL21-induced pDCs can directly suppress effector T cell function. We found that CCL21-induced pDCs significantly suppressed the proliferation of CD8+ T cells (Figure 5C). When co-cultured at a 1:2 ratio of pDCs to T cells for 3 days, CD8+ T cells decreased from 40% to 23%, and proliferation was suppressed by 25%. No significant suppression was observed when mDCs were co-cultured with CD8+ T cells. This suppression was reversible by either CCL21 or CCR7 antibody neutralization, suggesting a CCL21-specific effect. Interestingly, we did not find any effect of CCL21-induced pDCs on Treg differentiation or function (Figure 5D and Appendix A), which may indicate a complex role of the TME and its influence on the overall function of GBM-infiltrating pDCs compared to CCL21-induced pDCs in vitro (Figure 5E). We have previously published that pDC depletion in the diphtheria toxin receptor-mediated conditional knockout mouse model of GBM results in significantly increased overall survival of the tumor-bearing mice [5]. To further examine the future therapeutic potential of antibody-mediated targeting of CCL21 to decrease pDC tumor infiltration, we examined the effect of CCL21 blockade in vivo on pDC migration and activation. We found that after one week of CCL21 antibody administration, both pDC infiltration and activation were slightly decreased in the GBM-bearing brain (Appendix A). In our future work, we plan to optimize the dosing and therapeutic combination of CCL21 antibodies to explore this novel therapeutic option.

## 4. Discussion

Dendritic cells play an important role in linking innate and adaptive immunity and maintaining tolerance [49]. DCs are critical determinants of initiating and sustaining effective anti-tumor immune responses, which largely dictate the efficiency of immunotherapies. As an important subset of DCs, pDCs are known for their ability to secrete large amounts of IFN-α [1,50]. Depending on the environment, pDCs can be a double-edged sword, as they may contribute to tumor clearance by inducing anti-tumor immunity or tumor progression through immunosuppression [1,2,3]. Tumor-induced pDC expression of ICOSL, PD-L1, and CLTA-4 promotes the establishment of an immunosuppressive microenvironment through the suppression of T cell responses [1,51,52] and the activation of Tregs [53,54,55]. Tumor-infiltrating pDCs can also produce IDO, an enzyme that inhibits T cell proliferation, contributing to tumor immunosuppression [56,57]. Our previous data have demonstrated that pDC infiltration is associated with GBM tumor immunosuppression as well as a poor outcome, and selective pDC depletion increases the survival of tumor-bearing mice. However, the exact mechanism of how tumor cells interact with pDCs to influence their behavior in the context of GBM is not known. In this study, we demonstrate that GBM recruits pDCs to tumor sites via the CCL21-CCR7 signaling pathway. Moreover, these CCL21-induced pDCs show a regulatory function, directly suppressing CD8+ T cell proliferation and contributing to the immunosuppressive tumor microenvironment.

Chemokine signaling, as part of the tumor microenvironment, plays a vital role in tumorigenesis, proliferation, angiogenesis, tumor progression, and metastasis in many different types of cancer. The different chemokine signaling pathways relevant to GBM were reviewed in detail recently [58,59]. In brief, chemokine signaling is crucial for tumor angiogenesis (CXCR2 and CXCR4 [60]), immunosuppression (CXCR3 and CCR2 [13,35,61]), and tumor progression and metastasis (CCR5 and CXCR6 [61]). Recently, CCL21-CCR7 signaling was investigated as a potential anti-tumor target. It has been investigated in preclinical and clinical studies of breast, prostate, head and neck, and colon cancers, as well as B cell malignancies [19]. However, the literature on the role of CCL21-CCR7 signaling in brain tumors is limited. A recent study showed that lymphatic endothelial-like cells present in GBM could promote growth of CCR7-positive glioblastoma stem cells through CCL21 secretion [62]. It is also reported that CCL21-CCR7 signaling promotes microglia/macrophage recruitment and chemotherapy resistance in GBM [19]. On the other hand, CCL21-CCR7 signaling increases the lymph node homing of DCs, thus enhancing the effects of immunotherapy on brain tumors and metastases [63,64]. Therefore, the role of CCL21-CCR7 signaling in GBM is not yet fully understood.

Our data collectively highlight that CCL21 plays three critical roles in the GBM microenvironment. First, GBM cells specifically recruit pDCs to the tumor site through CCL21-CCR7 signaling. CCR7 and ACKR4 are the two most common receptors of CCL21. CCR7 has been intensively studied in different types of tumors, including GBM, and is negatively related to patient survival [18,19]. As one of the distinct atypical chemokine receptors, ACKR4 is also involved in the progression of malignant tumors. Although ACKR4 is expressed by cancer cells of various malignancies, including breast [65,66], liver [67], and colon cancer [68], it seems that this receptor plays a protective role against tumor development [59]. However, little is known about the role of ACKR4 in GBM. No current studies have assessed the expression and function of ACKR4 on immune cells, such as DCs. We found significant amounts of MHC-II and ACKR4 double-positive pDCs enriched in the GBM microenvironment. Due to the lack of an effective neutralized antibody, we cannot confirm if ACKR4 contributes to pDC recruitment together with CCR7. In accordance with the previously published literature [45], our data suggest that ACKR4 recruits β-arrestin and contributes to CCL21-driven intracellular signal transduction. Since the ACKR4 knockout mouse model has been established, future work may further demonstrate our findings.

Second, CCL21 induces pDC activation in the GBM TME. Among the infiltrated pDCs in GBM, more than 90% show the activated phenotype, represented by increased ACKR4 and MHC-II expression. In vitro experiments confirm that CCL21 can upregulate MHC-II transcription. MHC-II is tightly regulated by transcriptional activation involving a protein complex known as enhanceosome containing Regulatory factor x (RFX), cyclic AMP responsive-element-binding protein (CREB), Y-box specific nuclear transcription factor Y (NFY), and Class II major histocompatibility complex transactivator (CIITA) [38]. We further found that CIITA, a master control factor that regulates the expression of MHC II, is transcriptionally regulated by CCL21. Additionally, activated CIITA and β-arrestin, triggered by receptor activation, translocate to the nucleus and are selectively enriched at the specific MHC-II promoter, where it facilitates the recruitment of transcription co-factors and histone modifiers, resulting in transcription regulation. pDCs are produced in the bone marrow and emerge as mature cells in the periphery. Both pDCs and mDCs effectively migrate to the brain, LN, and spleen through either lymph nodes or the vasculature. We have previously shown that the frequency of pDCs in the LN increases at 1 week after tumor implantation in the GL261 mouse model of GBM but decreases by week 3 of tumor progression. This result suggests that the frequency of pDCs in the LN, BM, and blood might change dramatically during tumor progression due to migration and alteration in the TME. However, this hypothesis needs to be further explored.

Finally, CCL21 induces a regulatory phenotype in pDCs, likely contributing to overall GBM-mediated immunosuppression. Several studies have shown that in the context of different types of cancers, the immunosuppressive function of tumor-associated pDCs is associated with an increase in Tregs [53,54,56,57]. We examined if CCL21-induced pDCs suppress anti-tumor immunity via Tregs; however, using co-culture of CCL21-induced pDCs with naïve CD4 T cells, neither Treg lineage induction nor Treg proliferation was observed. This result suggests a critical role of the TME due to a difference between CCL21-induced pDCs and tumor-associated pDCs, which may closely interact with a diversity of cytokines/chemokines, including CCL21. Next, we explored the phenotype of CCL21-induced pDCs. Compared with non-treated pDCs, CCL21-induced pDCs upregulated the co-inhibitory factor PD-L1, which plays an important role in various malignancies where it can attenuate the host immune response to tumor cells [69]. Another immunomodulatory factor, IL10, was also increased in CCL21-induced pDCs. Increased IL-10 expression has been demonstrated to suppress anti-tumor immunity. Furthermore, the co-culture assay provides direct evidence that CCL21-induced pDCs suppressed the CD8 effector T cell proliferation. Taken together, all these data suggest that CCL21-induced pDCs show a regulatory phenotype that contributes to the tumor immunosuppressive microenvironment.

## 5. Conclusions

In summary, our study demonstrates that GBM cells recruit pDCs to tumor sites via CCL21-CCR7 signaling. The recruited pDCs have an activated and regulatory phenotype. This outlines a mechanism of how tumor cells regulate the immune microenvironment through CCL21. Moreover, we also found that either the CCL21 or CCR7 antibody could significantly neutralize the CCL21 effect, which provides a rationale for therapeutically targeting CCL21 as a potential novel therapeutic target for GBM. Further exploration and optimization of agents that target CCL21 might lead to viable clinical therapies.

## Figures and Tables

**Figure 1 cancers-16-03459-f001:**
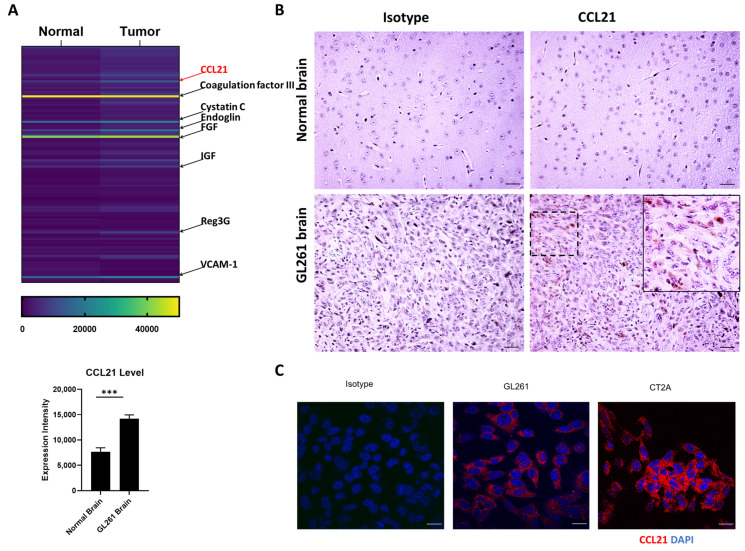
CCL21 is highly expressed by mouse GBM cells. (**A**) Heatmap of altered protein from GL261 tumor-bearing brain and normal brain by mouse cytokine/chemokine array. The density of the CCL21 protein level is analyzed by ImageJ (n = 3). Data represent mean ± SEM. *** *p* < 0.001. (**B**) Immunohistochemical (IHC) staining for CCL21 in GL261 tumor-bearing brain and normal brain. Scale bar = 50 μm. (**C**) Immunocytochemistry staining for intracellular CCL21 in GL261 and CT2A. Scale bar = 20 μm.

**Figure 2 cancers-16-03459-f002:**
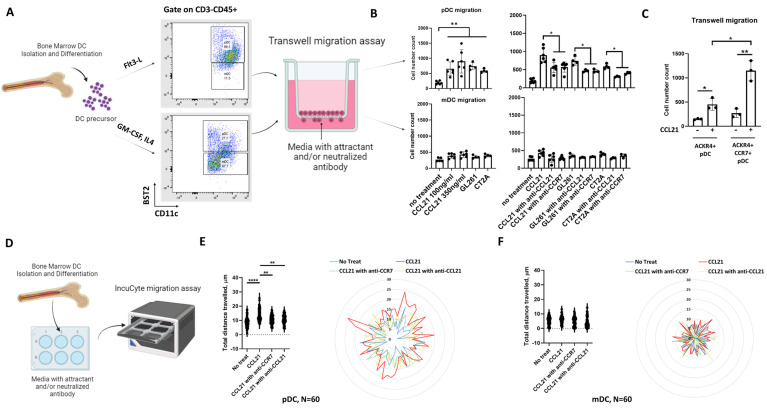
GBM mediates pDC, not mDC, migration via CCL21. (**A**) Schematic of transwell migration assays followed by flow cytometry. (**B**) Quantification of transwell migration assay of pDCs and mDCs in response to CCL21 protein, tumor cells, and neutralized antibodies. (**C**) Quantification of transwell migration assay of ACKR4+ or ACKR4+/CCR7+ sorted pDCs in response to CCL21 protein. (**D**) Schematic of IncuCyte migration assay. (**E**) Dot graph and radar plot depicting total distance traveled by pDCs in response to CCL21 protein and neutralized antibodies. (**F**) Dot graph and radar plot depicting total distance traveled by mDCs in response to CCL21 protein and neutralized antibodies. N = 60 cells, data represent mean ± SEM. * *p* < 0.05, ** *p* < 0.01, **** *p* < 0.0001.

**Figure 3 cancers-16-03459-f003:**
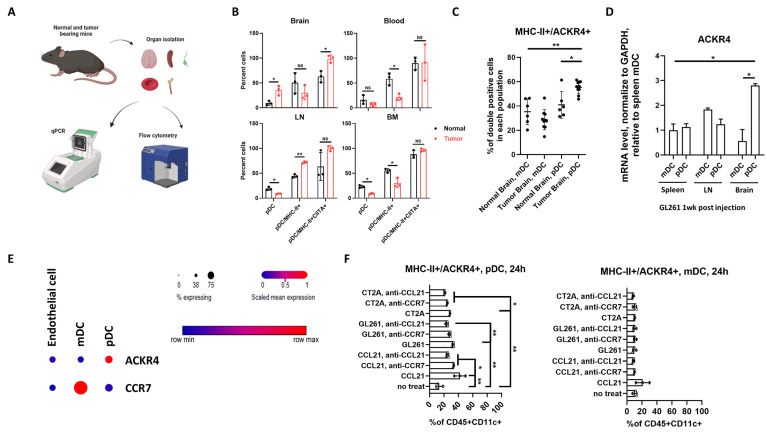
GBM mediates pDC activation via CCL21. (**A**) Schematic representation of tissue dissociation, qPCR, and flow cytometry experiments. (**B**) Distribution of pDCs, MHC-II+ pDCs, and MHC-II+/CIITA+ pDCs in different immune organs from normal and tumor-bearing mice. (**C**) Increased MHC-II+/ACKR4+ pDCs were found in tumor-bearing brains, compared with normal brains or mDCs in tumor-bearing brains. (**D**) ACKR4 gene expression in pDCs and mDCs sorted from the spleen, LN, and brain of GL261-bearing mice (n = 4). (**E**) Single-nucleus RNA sequencing data in cervical cancer. The figure is exported from Single Cell Portal, BROAD [36,37]. (**F**) Activation of pDCs, not mDCs, in response to CCL21 protein, tumor cell co-culture, and neutralized antibodies. n ≥ 3, data represent mean ± SEM. NS no significant, * *p* < 0.05, ** *p* < 0.01.

**Figure 4 cancers-16-03459-f004:**
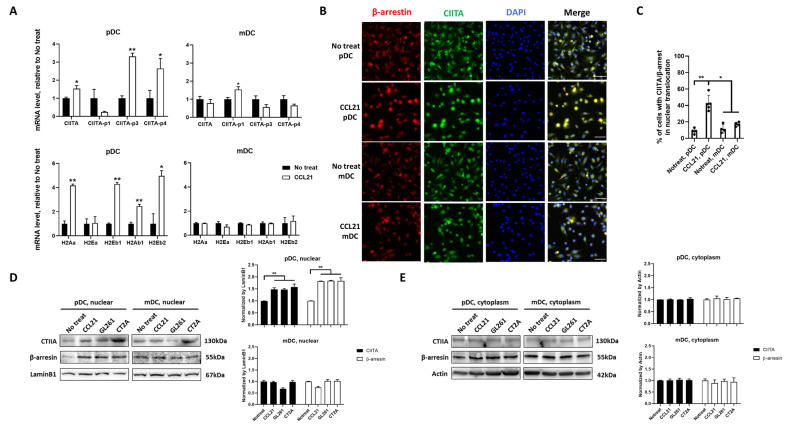
CCL21 induces pDC activation through the CCR7/ACKR4—β-arrestin/CIITA pathway. (**A**) qPCR was used to quantify gene expression and is reported as fold change normalized to untreated controls. (**B**) Representative immunostaining images of β-arrestin and CIITA and (**C**) quantification of the immunostaining images. Nuclear translocation was quantified by counting positive cell numbers. Scale bar = 50 μm. (**D**) Immunoblot analysis of β-arrestin and CIITA proteins in nuclear and cytoplasm fractions of pDCs and (**E**) immunoblot analysis of β-arrestin and CIITA proteins in nuclear and cytoplasm fractions of mDCs. Quantification: normalization to actin/LaminB1. Data represent mean ± SEM. * *p* < 0.05, ** *p* < 0.01.

**Figure 5 cancers-16-03459-f005:**
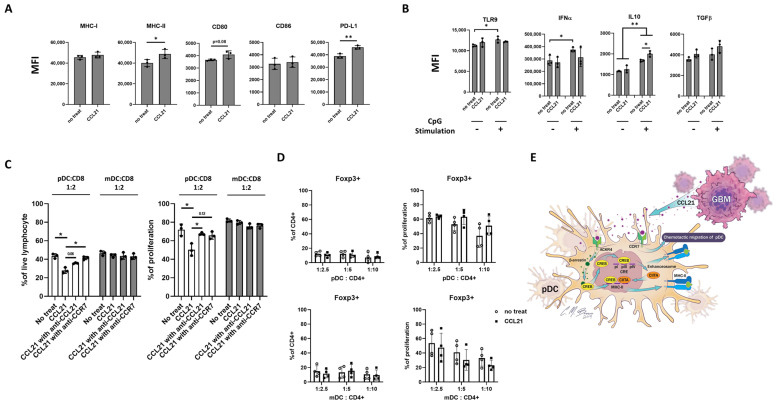
CCL21-induced pDCs show a regulatory phenotype. (**A**) Phenotypic analysis of pDCs with CCL21 treatment. (**B**) Flow cytometry analysis of TLR9 expression and cytokine secretion of CCL21-pretreated pDCs under CpG stimulation. (**C**) CD8+ T cells were co-cultured with pretreated pDCs. T cell population and proliferation were analyzed and quantified by flow cytometry. (**D**) CD4+ naïve T cells were co-cultured with pretreated pDCs. Foxp3+ T cell population and proliferation were analyzed and quantified by flow cytometry. (**E**) Mechanistic overview of CCL21 induction of pDCs. Data represent mean ± SEM. * *p* < 0.05, ** *p* < 0.01.

## Data Availability

The original contributions presented in the study are included in the article/Appendix A, further inquiries can be directed to the corresponding author.

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
