# Peer review of "CCL21 Induces Plasmacytoid Dendritic Cell Migration and Activation in a Mouse Model of Glioblastoma"

_cancers, 2024, doi:10.3390/cancers16203459_

Round 1

Reviewer 1 Report

Comments and Suggestions for Authors

The authors investigate the effect of glioblastoma cellular activity on plasmacytoid dendritic cells migration and activation activity via using immunocompetent mouse models. The study is well designed, attractive and ready for publishing after some minor change. The following are my minor comments.

1)     In Figure 1 (B), the IHC image is too small, except the image CCL21 stained GL261 brain IHC imaged. Please include additional x20 (or x40) magnification image, such as the image CCL21 stained GL261 IHC image.

a.      Similar comment on Figure 4 (B)

2)     Figure 2 is too small. Please enlarge and rearrange the images accordingly.

a.      Similar comment on Figure 5 (E).

Author Response

We thank all the reviewers and the editor for their time and appreciate their constructive feedback. We have reviewed all the suggestions from the reviewers and have addressed all their concerns individually as outlined below.

Reviewer 1

The authors investigate the effect of glioblastoma cellular activity on plasmacytoid dendritic cells migration and activation activity via using immunocompetent mouse models. The study is well designed, attractive and ready for publishing after some minor change. The following are my minor comments.

1)     In Figure 1 (B), the IHC image is too small, except the image CCL21 stained GL261 brain IHC imaged. Please include additional x20 (or x40) magnification image, such as the image CCL21 stained GL261 IHC image.

  1. Similar comment on Figure 4 (B)

Response:

Figure quality is enhanced in the revised manuscript. Please refer to the individually uploaded high resolution Figure 1 and Figure 4 (B) respectively.

2)     Figure 2 is too small. Please enlarge and rearrange the images accordingly.

  1. Similar comment on Figure 5 (E).

Response:

Figure quality is enhanced in the revised manuscript. Please refer to the individually uploaded high resolution Figure 2 and Figure 5 (E) respectively.

Reviewer 2 Report

Comments and Suggestions for Authors

The manuscript by Zhao et al., titled "CCL21 Induces Plasmacytoid Dendritic Cells Migration and Activation in Mouse Model of Glioblastoma," makes important contributions by exploring the role of the chemokine CCL21 in the recruitment and activation of plasmacytoid dendritic cells (pDCs) in the glioblastoma (GBM) microenvironment. However, there are some points that warrant consideration for improving the study:

  1. While the focus on the CCL21-CCR7 pathway is relevant, it would be beneficial to include comparisons with other chemokine pathways to provide a broader context for the importance of this specific pathway in GBM.
  2. Including more detailed negative controls in the neutralization experiments could strengthen the confidence in the results, ensuring that the observed effects are indeed due to the inhibition of CCL21 and CCR7.
  3. The study interestingly addresses the role of pDCs induced by CCL21. However, it would be helpful to expand the discussion on discrepancies with previous studies that associate pDCs with Treg expansion, clarifying possible explanations for these findings.
  4. The conclusions on the potential neutralization of CCL21 as a therapy are promising. However, further exploration of long-term effects in the animal model, such as tumor progression, could provide additional support for these claims.
  5. The statistical analysis is well-conducted, but the inclusion of multiple comparison tests in certain analyses, such as CCL21 concentrations, could increase the robustness of the results.
  6. Including comparisons with other types of brain tumors would help demonstrate the specificity of the CCL21-CCR7 pathway in glioblastoma, enhancing the relevance of the findings.
  7. The conclusions on the therapeutic use of CCL21 neutralization are interesting, but it would be ideal to include more complete long-term experimental data to better support this perspective.
  8. The absence of multiple comparison tests, such as Bonferroni or Tukey, in analyses with different CCL21 concentrations could lead to misleading interpretations. Applying these tests would ensure greater rigor in data analysis, and it would be advisable to consider including them.
  9. The manuscript needs to indicate the source of cell types, as well as the products, equipment, brands, and company names used.
  10. I recommend increasing the font size of the figures and reorganizing them clearly, as it is currently difficult to observe the details of the information.
Comments on the Quality of English Language

It is recommended that the manuscript undergoes minor revisions to improve the clarity and fluency of the English language. While the content is generally well-written, refining certain phrases and correcting minor grammatical issues would enhance the overall readability and presentation

Author Response

We thank all the reviewers and the editor for their time and appreciate their constructive feedback. We have reviewed all the suggestions from the reviewers and have addressed all their concerns individually as outlined below.

Reviewer 2

The manuscript by Zhao et al., titled "CCL21 Induces Plasmacytoid Dendritic Cells Migration and Activation in Mouse Model of Glioblastoma," makes important contributions by exploring the role of the chemokine CCL21 in the recruitment and activation of plasmacytoid dendritic cells (pDCs) in the glioblastoma (GBM) microenvironment. However, there are some points that warrant consideration for improving the study:

  1. While the focus on the CCL21-CCR7 pathway is relevant, it would be beneficial to include comparisons with other chemokine pathways to provide a broader context for the importance of this specific pathway in GBM.

Response:

Issue addressed. Please refer to line 411 to 416 in the revised manuscript.

  1. Including more detailed negative controls in the neutralization experiments could strengthen the confidence in the results, ensuring that the observed effects are indeed due to the inhibition of CCL21 and CCR7.

Response:

In the current study, both anti-CCL21 and anti-CCR7 antibodies decrease pDCs migration and activation in response to both CCL21 stimulation and tumor co-culture conditions, which indicates that the observed effects are due to the blockade of CCL21-CCR7 signaling. Both antibodies are specific and have been verified in previous publications [1-4].

Reference

  1. St John AL, Abraham SN. Salmonella disrupts lymph node architecture by TLR4-mediated suppression of homeostatic chemokines. Nat Med. 2009 Nov;15(11):1259-65.
  2. Liu C, Ueno T, Kuse S, Saito F, Nitta T, Piali L, Nakano H, Kakiuchi T, Lipp M, Hollander GA, Takahama Y. The role of CCL21 in recruitment of T-precursor cells to fetal thymi. Blood. 2005 Jan 1;105(1):31-9.
  3. Mao FY, Lv YP, Hao CJ, Teng YS, Liu YG, Cheng P, Yang SM, Chen W, Liu T, Zou QM, Xie R, Xu JY, Zhuang Y. Helicobacter pylori-Induced Rev-erbα Fosters Gastric Bacteria Colonization by Impairing Host Innate and Adaptive Defense. Cell Mol Gastroenterol Hepatol. 2021;12(2):395-425.
  4. Pang MF, Georgoudaki AM, Lambut L, Johansson J, Tabor V, Hagikura K, Jin Y, Jansson M, Alexander JS, Nelson CM, Jakobsson L, Betsholtz C, Sund M, Karlsson MC, Fuxe J. TGF-β1-induced EMT promotes targeted migration of breast cancer cells through the lymphatic system by the activation of CCR7/CCL21-mediated chemotaxis. Oncogene. 2016 Feb 11;35(6):748-60.

  1. The study interestingly addresses the role of pDCs induced by CCL21. However, it would be helpful to expand the discussion on discrepancies with previous studies that associate pDCs with Treg expansion, clarifying possible explanations for these findings.

Response:

Issue addressed. Please refer to line 396-406 and line 463-479 in the revised manuscript.

  1. The conclusions on the potential neutralization of CCL21 as a therapy are promising. However, further exploration of long-term effects in the animal model, such as tumor progression, could provide additional support for these claims.
  2. The conclusions on the therapeutic use of CCL21 neutralization are interesting, but it would be ideal to include more complete long-term experimental data to better support this perspective.

Response:

We agree with the reviewer and are also every excited to explore the therapeutic implication of CC21 blockade in long term experiments. In our current experimental setting, CCL21 blockade decrease both pDC infiltration and activation to some extent, which indicates that this might have clinical benefit for GBM treatment. However optimal dose and frequency of dosing to get the desired pharmaceutical advantage needs to be extensively studied in pharmacokinetic and pharmacodynamic studies, which are out of the scope of this publication. In our future study, we will modify the antibody dosing and try different administration strategy to further address this issue and develop this novel treatment option.

  1. The statistical analysis is well-conducted, but the inclusion of multiple comparison tests in certain analyses, such as CCL21 concentrations, could increase the robustness of the results.
  2. The absence of multiple comparison tests, such as Bonferroni or Tukey, in analyses with different CCL21 concentrations could lead to misleading interpretations. Applying these tests would ensure greater rigor in data analysis, and it would be advisable to consider including them.

Response:

Issue addressed please refer to Method section line 180 to 184 in the revised manuscript.

  1. Including comparisons with other types of brain tumors would help demonstrate the specificity of the CCL21-CCR7 pathway in glioblastoma, enhancing the relevance of the findings.

Response:

Issue addressed. Please refer to line 416 to 426 in the revised manuscript.

  1. The manuscript needs to indicate the source of cell types, as well as the products, equipment, brands, and company names used.

Response:

Issue addressed. Please refer to Method section line 88 to 89 in the revised manuscript and Supplemental Materials.

  1. I recommend increasing the font size of the figures and reorganizing them clearly, as it is currently difficult to observe the details of the information.

Response:

Figure quality is enhanced in the revised manuscript. Please refer to the individually uploaded high resolution figures.

Reviewer 3 Report

Comments and Suggestions for Authors

In this work, the author discusses how GBM cells recruit pDCs to tumor locations and modify their function, ultimately contributing to the immunosuppressive TME. It is vital to handle the following concerns: The author should use CCR7 and ACRK4 knockdown cells to track pDC migration and activation through CCL21. To better understand the role of CCL21-induced pDCs in antitumor immunosuppression, the author should perform a differential gene expression analysis of pDCs with and without CCL21.

Comments on the Quality of English Language

There is no issue with the writing. 

Author Response

We thank all the reviewers and the editor for their time and appreciate their constructive feedback. We have reviewed all the suggestions from the reviewers and have addressed all their concerns individually as outlined below.

Reviewer 3

In this work, the author discusses how GBM cells recruit pDCs to tumor locations and modify their function, ultimately contributing to the immunosuppressive TME. It is vital to handle the following concerns: 

The author should use CCR7 and ACRK4 knockdown cells to track pDC migration and activation through CCL21.

Response:

Our overall goal is to develop novel therapeutic options for brain tumor patients. Although gene knockdown is a commonly used basic science technique for studying function of various genes, it is not an effective translational tool. Hence, we used specific blocking antibodies that can also be used as a therapeutic strategy for in vivo testing. Antibodies used in our studies are specific and has all the proper controls for robust validation of our findings. In our future study, we plan to use CCR7 and ACKR4 knockdown method to further explore the mechanism of CCL21 in GBM.

To better understand the role of CCL21-induced pDCs in antitumor immunosuppression, the author should perform a differential gene expression analysis of pDCs with and without CCL21.

Response:

Thanks for the suggestion. We agree with the reviewer that unbiased whole transcriptome analysis of pDCs in the presence or absence of CCL21 will be a great method to study all the possible mechanism of pDC mediated immunosuppression in the tumor microenvironment. In this manuscript we focused on studying the mechanism of pDC accumulation and activation in the tumor microenvironment. In our future study, we plan to perform the RNA sequencing study for further investigation of the mechanism of pDC mediated immunosuppression.

Round 2

Reviewer 2 Report

Comments and Suggestions for Authors

I believe the authors have satisfactorily addressed all the questions raised during the review. The responses provided were clear and well-founded, addressing the critical points of the manuscript. Therefore, I recommend the publication of the work

Comments on the Quality of English Language

The quality of the English language in the manuscript is adequate, with clear and coherent sentences that effectively convey the intended meaning. There are no significant issues regarding grammar, spelling, or punctuation